# Surrogate-Based Quantification of Policy Uncertainty in Generative Flow Networks

## Abstract

Generative flow networks are able to sample, via sequential construction, high-reward, complex objects according to a reward function. However, such reward functions are often estimated approximately from noisy data, leading to epistemic uncertainty in the learnt policy. We present an approach to quantify this uncertainty by constructing a surrogate model composed of a polynomial chaos expansion, fit on a small ensemble of trained flow networks. This model learns the relationship between reward functions, parametrised in a low-dimensional space, and the probability distributions over actions at each step along a trajectory of the flow network. The surrogate model can then be used for inexpensive Monte Carlo sampling to estimate the uncertainty in the policy given uncertain rewards. We illustrate the performance of our approach on a discrete and continuous grid-world, symbolic regression, and a Bayesian structure learning task.

## 1 Introduction

*Generative* artifical intelligence (AI) describes the subclass of techniques which learn a probabilistic model of the data-generating process. This yields advantages such as the ability to sample new data and to estimate the confidence in a model's output. Generative architectures such as generative flow networks (GFNs)[1] [Bengio et al. (2021)], transformers [Vaswani et al. (2017)], variational autoencoders (VAEs) [Kingma & Welling (2013)], diffusion models [Sohl-Dickstein et al. (2015)], and generative adversarial networks (GANs) [Goodfellow et al. (2014)] have become state-of-the-art approaches with a host of traditional and blue-sky applications. Due to the probabilistic nature of such models, it is common to describe them as 'uncertainty aware', as they are typically able to compute the likelihood of each sample. It is important to note that this *aleatoric uncertainty* in a model's prediction or output is that which stems from the probabilistic process that the model has learnt, but assumes that the generative model is precise and exactly correct. However, models learn a generative process from finite data or through self-play, via stochastic optimisation. This implies that an additional form of uncertainty at the level of the model itself, so-called *epistemic uncertainty*, can arise, as a variety of generating processes can be learnt given alternative data or realisations of a stochastic input [Gal (2016)]. Quantifying this model uncertainty is key to understanding the capabilities and limitations of a given generative model and its predictions [Chakraborti et al. (2025)]. Two well-known deep learning paradigms for quantifying such uncertainty are the use of Bayesian neural networks, where network weights are random variables whose posterior estimates are updated during training [Blundell et al. (2015); MacKay (1995)], and Monte-Carlo (MC) sampling/dropout, where training an ensemble of identical models with stochastic variation in input data yields an empirical distribution in the space of generative processes [Gal & Ghahramani (2016); Lakshminarayanan et al. (2017)]. However, Bayesian methods are complicated to implement and do not scale to large models, whereas for the MC approach, repeated training is computationally expensive and prohibitively slow.

For an alternative approach, we turn to an analogous situation that is common in a variety of computational engineering problems with stochastic input [Conti et al. (2024)], where repeated, high-fidelity simulations are computationally intractable, yet an understanding of the output uncertainty is desired. Such problems have led to the development of the field of uncertainty quantification (UQ)

---

[1]Also known as *GFlowNets*.

[Sullivan (2015)]. A cornerstone approach in UQ is to construct computationally-efficient 'surrogate' models using an array of techniques such as polynomial chaos expansions (PCEs) [Wiener (1938); Xiu & Karniadakis (2002)], machine learning (ML) models [Conti et al. (2024); Vlachas et al. (2022)], Gaussian processes [Rasmussen & Williams (2006)], or a combination of these approaches [Shustin et al. (2022)]. Typically, these methods approximate an input-output relationship by fitting to a limited number of examples obtained from the high-fidelity, 'black-box' model. Finally, performing MC sampling from the surrogate model allows model uncertainty to be quantified efficiently [Sullivan (2015)]. Similarly, the training procedure of a large generative AI model functions as a black-box. Repeated training of models with different input data, spanning the full distribution, is akin to MC sampling from a high-fidelity model, and therefore is computationally prohibitive. To address this problem, we investigate surrogate-modelling approaches for generative models, focusing on GFNs in situations where the reward is uncertain, thus producing uncertainty in the learnt policy. This paper is structured as follows. We present a brief introduction to GFNs with uncertain rewards, as well as to PCE. We then illustrate the performance of our approach on various example problems, including discrete and continuous grid-worlds, symbolic regression, and a Bayesian structure learning task.

## 2 GENERATIVE FLOW NETWORKS WITH UNCERTAIN REWARDS

GFNs are a novel and powerful architecture at the interface between generative and reinforcement learning, and have been used for generating molecular structures [Bengio et al. (2021)], solving combinatorial optimisation problems [Zhang et al. (2023)], Bayesian structure learning [Deleu et al. (2022)], and biological sequence design [Jain et al. (2022)]. Variants exist for both continuous reward functions [Lahlou et al. (2023)] and stochastic transition graphs [Pan et al. (2023)]. A GFN learns a policy for iteratively constructing a trajectory in the flow network according to a given reward function. Whilst the policy encodes the probability of each transition through the flow network, a GFN is unable to express the level of uncertainty in the policy itself. In particular, in many applications, such as molecular candidate sampling or Bayesian structure learning, the reward function is estimated from noisy experimental data, and/or expressed by a neural network (NN) [Bengio et al. (2021); Deleu et al. (2022)]. As a result, there is inherent uncertainty in the reward function estimation, leading to epistemic uncertainty in the learnt policy that GFNs fail to express. Whilst there have been some attempts to encode such uncertainty by considering a distribution of GFNs [Jain et al. (2022); Zhang et al. (2024); Liu et al. (2023)], this remains an open problem.

### 2.1 FLOW NETWORKS

A *flow network* is defined by a pair $(G, F)$. Here $G = (\mathcal{S}, \mathcal{E})$ is a directed acyclic graph (DAG) with a finite number of vertices, which each represent a state $s \in \mathcal{S}$, and where each edge $e = (s \rightarrow s') \in \mathcal{E}$ is a directed connection representing a possible transition between states. The *flow function* $F : \mathcal{T} \rightarrow \mathbb{R}^+$ is a positively-valued function, defined on $\mathcal{T}$, the set of all trajectories through the state-space DAG. For a function $F$ to define a valid flow network, one can derive that it must satisfy the *flow matching condition*,

$$F(s) = \sum_{(s'' \rightarrow s) \in \mathcal{E}} F(s'' \rightarrow s) = \sum_{(s \rightarrow s') \in \mathcal{E}} F(s \rightarrow s'), \tag{1}$$

for all $s \in \mathcal{S}$, which ensures that the flow is 'consistent'. The set of states for which there are no possible forward transitions is denoted *terminating states*, $\mathcal{S}_f \subseteq \mathcal{S}$. Assuming that our flow network is *Markovian*[2], a valid flow function yields a stochastic process over $\mathcal{S}$ with forward and backward transition probabilities given by,

$$P_f(s'|s) = \frac{F(s \rightarrow s')}{\sum_{s'':(s \rightarrow s'') \in \mathcal{E}} F(s \rightarrow s'')}, \quad P_b(s'|s) = \frac{F(s' \rightarrow s)}{\sum_{s'':(s'' \rightarrow s) \in \mathcal{E}} F(s'' \rightarrow s)}, \tag{2}$$

which, given the state $s$, are the probabilities of transitioning to state $s'$ forwards, $P_f$, or backwards, $P_b$, respectively. Finally, we define $P_T(s)$ to be the *terminating* probability of each terminating

---

[2]Given the state $s$, *Markovianity* implies that each transition in our flow network is sampled stochastically according to the relative flow along each edge $s \rightarrow s'$, independent of the previous states along the trajectory [Bengio et al. (2023)].

state $s \in \mathcal{S}_f$, i.e. the probability of a trajectory ending in that terminating state. Flow networks are particularly useful in the case where we have a *reward function*, $R : \mathcal{S}_f \to \mathbb{R}^+$, defined on the terminating states. Given some total amount of flow in the network, $Z = \sum_{s \in \mathcal{S}_f} R(s)$, and the constraint that $F(s) = R(s)$ for $s \in \mathcal{S}_f$, then $P_T(s) = R(s)/Z \propto R(s)$ for $s \in \mathcal{S}_f$, i.e. a valid flow will sample terminating states in proportion to their relative reward. This property is true for any valid flow, which may not be unique. As a result, flow networks become an attractive method for sampling high-reward candidates from the set $\mathcal{S}_f$. Given a DAG and a target reward function, $R(s)$, constructing a valid flow is a non-trivial problem. Instead, we use a NN, which acts as a function approximator, and seek to minimise the error in some flow consistency condition. For details on the loss functions associated with these conditions, see App. A.1.

## 2.2 CONTINUOUS GENERATIVE FLOW NETWORKS

GFNs can be extended to continuous state-spaces, offering a host of novel applications. The theory of *continuous GFN* (cGFN) naturally, but non-trivially, generalises the definitions in Sec. 2.1. Here we will introduce only a minimal theory of a cGFN which is sufficient for a continuous 'grid'[3] environment that an agent explores. For a more general mathematical theory, we point the reader to Lahlou et al. (2023). We consider a continuous, measurable domain $\Omega$, where any state $\omega \in \Omega$ can be both an intermediate or terminating state and where each state is reachable from all others. Therefore, a reward function $R : \Omega \to \mathbb{R}^+$ is defined on the whole domain. Next, we define a forward and backward policy, $P_f(\omega \to \omega')$ and $P_b(\omega \to \omega')$, which represents the density of a transition from $\omega \to \omega'$ for every pair $\omega, \omega' \in \Omega$, forwards or backwards, respectively. Given the pair $F = (\mu, P_f)$, where $\mu$ is a measure defined on $\Omega$, the flow matching condition is met if for any bounded measurable function $u : \Omega \to \mathbb{R}$ that satisfies $u(s_0) = 0$, we have that,

$$\int_\Omega u(\omega')\mu(d\omega') = \iint_{\Omega^2} u(\omega')\mu(d\omega)P_f(\omega \to \omega')d\omega', \tag{3}$$

where $F$ is said to be a flow. The reward matching condition is then given by,

$$R(\omega) = \int_\Omega P_f(\omega' \to \omega)\mu(d\omega'). \tag{4}$$

## 2.3 UNCERTAIN REWARDS AND UNCERTAIN POLICIES

We focus on the situation of an uncertain reward function, $R$, sampled from a distribution, $R \sim \mathcal{R}(\mathbb{E}[R])$, where $\mathbb{E}[R]$ is the true (expected) reward. This variation can arise from measurement error or epistemic uncertainty in the reward itself, with examples such as rewards which are parametrised by NNs, or those calculated from noisy experimental data. A GFN trained on a sampled reward function learns the flow $F_\theta(\cdot|R)$ which is a sample from the distribution over possible flow functions given a particular architecture and training objective. The goal of UQ in this scenario is to build up a picture of the marginal distribution over policies obtained by integrating over $\mathcal{R}$, for a fixed architecture and objective. The space of all possible trajectories is prohibitively large; thus, we focus on the policy *along a trajectory*. Given a trajectory, $\tau = s_0 \to ... \to s_n$, sampled from a GFN (or an ensemble of GFNs), we are interested in the distribution and expected value of the policy along that trajectory, which defines a collection of random variables (RV) $\{P_\theta(\cdot|s_t), t = 0, 1, ..., n-1\}$, where each is a distribution over the set of states that are reachable in one step from $s_t$. In the case of the cGFN, this is a probability density function (PDF) defined by a set of parameters, i.e. mean and variance for a Gaussian policy. We say that the distribution $P(\cdot|s_t)$ is a sample from the distribution over policies $\mathcal{P}(s_t)$.

## 3 SURROGATE MODELLING WITH POLYNOMIAL CHAOS EXPANSIONS

In order to perform surrogate modelling, we assume that there is a mapping $\Lambda_t : \text{supp}(\mathcal{R}) \to \text{supp}(\mathcal{P}(s_t))$, between the set of all reward functions and set of possible policies at each step in the trajectory. We approximate $\Lambda_t$ with a flexible model that is easy to fit using only a few realisations,

---

[3]We call this a 'grid world' to remain consistent with the literature in discrete settings [Bengio et al. (2021)]. However, we are referring to a continuous domain.

and which allows for fast sampling, eschewing the need to retrain a great number of GFNs. However, $\mathrm{supp}(\mathcal{R})$ is a space of functions, which may be high-dimensional and challenging to parametrise. When an obvious parameterisation is not available, we must learn a latent representation of the input space using dimensionality reduction such as principal component analysis (PCA), or for nonlinear data, a VAE. Moreover, the VAE is able to project complex, high-dimensional distributions to a more Gaussian representation. This defines a mapping $\phi : \Gamma \to \mathrm{supp}(\mathcal{R})$ where $\phi(\boldsymbol{\mu}) = R$ and $\boldsymbol{\mu} \in \Gamma$ is a (low-dimensional) set of parameters that represent the reward function. The surrogate model then learns the mapping $\Lambda_t \circ \phi(\boldsymbol{\mu}) = P(\cdot|s_t)$, which maps between the input space $\Gamma$ where each point represents a reward function, and the set of policies at step $s_t$.

### 3.1 POLYNOMIAL CHAOS EXPANSIONS

PCEs are flexible and inexpensive surrogate models for approximating input-output data [Wiener (1938); Xiu & Karniadakis (2002)]. A random variable, $Y \in \mathbb{R}$, with finite variance, can be expressed as a polynomial function of a random vector, $\boldsymbol{X} \in \mathbb{R}^m$,

$$Y = \sum_{\mathbf{j} \in \mathbb{N}^m} c_{\mathbf{j}} \varphi_{\mathbf{j}}(\boldsymbol{X}), \tag{5}$$

where $c_{\mathbf{j}}$ are coefficients and $\varphi_{\mathbf{j}}$ form an orthonormal basis of polynomials (see App. B). For typical distributions, such as Gaussian or uniform, the set of orthonormal polynomials that achieves optimal convergence is well known (Hermite and Legendre polynomials, respectively [Xiu & Karniadakis (2002)]). Given input-output data $(\mathbf{X}, \mathbf{Y})$, where $\mathbf{X} = \{\mathbf{x}_1, ..., \mathbf{x}_n\}$ with $\mathbf{x}_i \in \mathbb{R}^m$ and $\mathbf{Y} = (y_1, ..., y_n)$ with $y_i \in \mathbb{R}$, a surrogate model can be constructed by estimating the coefficients $c_{\mathbf{j}}$ which best fit this data. A simple approach to fitting these coefficients is to perform (regularised) *regression* [Hastie et al. (2009)]. In the case that the output is multidimensional, i.e. $\mathbf{Y} = (\mathbf{y_1}, ..., \mathbf{y_n})$, then we can fit a PCE to each output variable in turn. When the output is a discrete probability distribution, we first apply the logit transformation to outputs before fitting the PCE as polynomial functions are unbounded (see App. B.1). We then transform this back into a distribution using the soft-max [Goodfellow et al. (2016)].

### 3.2 INTEGRATING PCEs INTO GFNs

We begin by training a distinct 'training' and 'testing' ensemble of GFNs on the same task, each with a stochastically sampled reward function. Given a trajectory of interest, $\{s_0, ..., s_n\}$, at each step $s_t$, we extract the policy, a distribution over actions, from each GFN in the training ensemble, labelled as $l \in \{1, ..., L\}$. Given the low-dimensional representation of each reward function, $\boldsymbol{\mu}_l$, from a known distribution, we fit a distinct PCE, using the orthonormal basis associated with the distribution[4]. For each action and step, we find the set of coefficients, $c_{\mathbf{j}}$, by optimising

$$c_{\mathbf{j}} = \mathrm{argmin}_{\tilde{c}_{\mathbf{j}}} \sum_{l=1}^{L} \left\| \sum_{\mathbf{j} \in \Theta} \tilde{c}_{\mathbf{j}} \varphi_{\mathbf{j}}(\boldsymbol{\mu}_l) - \mathrm{logit}(p_k^l(s_t)) \right\|^2, \tag{6}$$

where $p_k^l(s_t)$ is the probability of action $k$ at state $s_t$ from model $l$ in the training ensemble, and $\Theta$ are the truncated indices of the expansion. Sampling new inputs from the low-dimensional reward space, we can use the PCE to sample surrogate policies along this trajectory. We compare these samples to the testing ensemble, which was not used to fit the PCE.

## 4 NUMERICAL EXPERIMENTS

In this section, we focus on implementations of the UQ framework applied to a number of example problems; specifically, a discrete and continuous grid-world, symbolic regression, and Bayesian structure learning.

### 4.1 DISCRETE GRID-WORLD WITH UNCERTAIN REWARDS

We begin with a $10 \times 10$ grid where each square is assigned a low, mid, or high reward ($R(x) = 0.1$, 40 or 200, respectively), as shown in Panel $a$), Fig. 1. Starting at a random initial position, the GFN

---

[4]For example, when using the $\beta$-VAE, we assume the latent space is a Gaussian distribution.

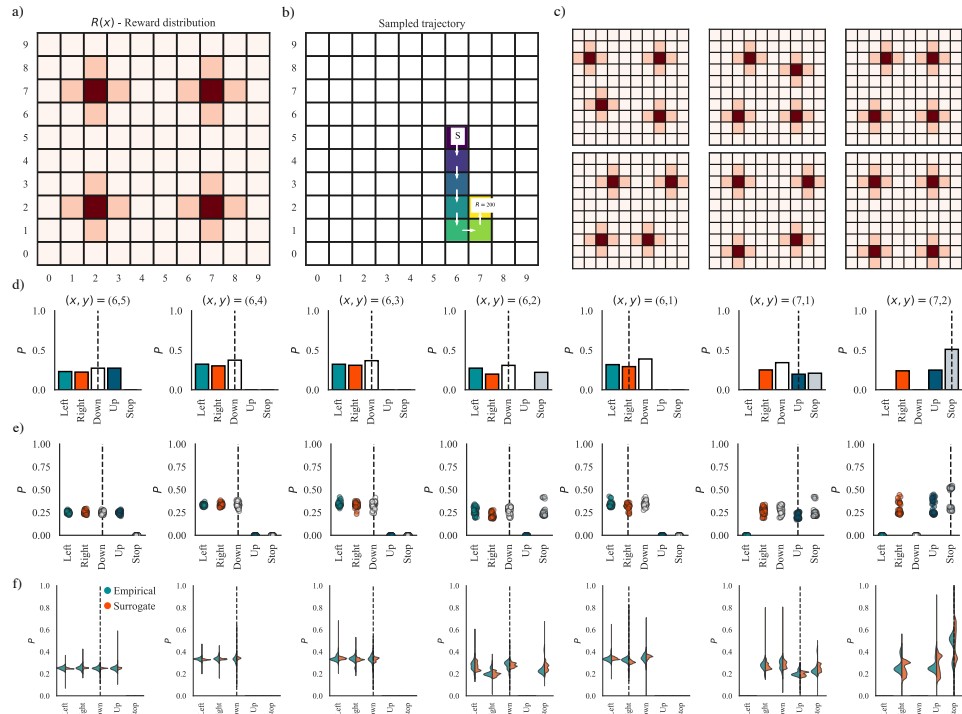

Figure 1: **Discrete grid-world with uncertain rewards**. $a$) Ground-truth reward function defined on $10 \times 10$ grid. $b$) Example trajectory terminating at a high-reward location. $c$) Random samples from the distribution over possible reward functions. $d$) Policy along the trajectory from $b$) for a single model. $e$) Distribution over policies from an ensemble of 50 models trained on sampled variations of the reward grid. $f$) Empirical testing ensemble of models compared with outputs of PCE surrogate model. The PCE model can capture the shape of the distribution, including bimodality.

learns to navigate the grid by moving up, down, left and right, without being able to move through boundaries or go back on itself[5] (Panel $b$), Fig. 1). The policy can terminate after any number of steps, up to a maximum, and the reward is given by the final position. Trajectories may only terminate if they reach a mid or high reward, or if they get trapped. To stochastically vary the reward function, we begin with the ground-truth reward grid and for each of the four $+$, we sample, with probability $p$, whether or not it will shift. If the $+$ is chosen to shift, we choose, uniformly, whether it will shift up, down, left or right, which yields $5^4$ possible reward functions. Panel $c$) of Fig. 1 shows six random samples of a reward grid. We train two ensembles, 50 training models and 100 testing models, using random samples from the distribution over reward functions. For details on the architecture and training objective see App. E.

**Uncertain policy along a trajectory.** The policy at each step along a trajectory is a discrete probability distribution over the actions ['Left', 'Right', 'Up', 'Down', 'Stop']. Panels $d$) and $e$) of Fig. 1 show the policy over the trajectory from Panel $b$), for a single model, and the distribution of policies for an ensemble of models, respectively. However, as the ensemble is only 50 samples, this distribution does not capture the complete shape or variance of the distribution over the policy.

**Low-dimensional parameterisation of reward functions.** In this example, the reward function is specified by a $10 \times 10$ array of numbers. This is a high-dimensional representation that contains much redundant information. Following the discussion in Sec. 2.3, we learn a low-dimensional representation of our reward function space using a VAE. Using a one-hot encoding of the reward grid ($3 \times 10 \times 10$), we train a $\beta$−VAE [Higgins et al. (2017)] to project each grid to a latent Gaussian distribution with $d = 2$ independent components. By increasing $\beta$, we increase the strength of the prior assumption that the latent space is distributed as $\mathcal{N}(0, 1)^d$, which can increase *disentanglement*

---

[5]This condition prevents cycles. Nevertheless, these can be avoided using an augmented state-space with a time-stamp [Bengio et al. (2023)].

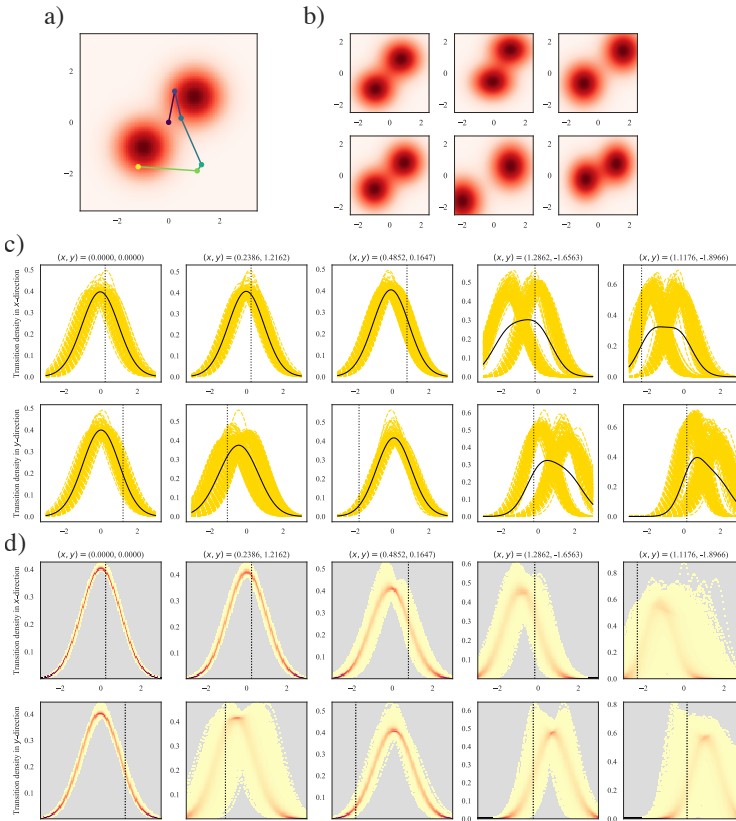

Figure 2: **Continuous grid-world with uncertain rewards.** $a$) Ground-truth reward function with an example trajectory of 5 steps. $b$) Samples of uncertain reward functions from a distribution. $c$) Policies along the trajectory from Panel $a$) in $x-$ and $y-$ direction (columns) for each of 5 steps (rows). Policies are taken from a testing ensemble of 100 GFNs trained on samples from the reward distribution. The black line is the mean policy. $d$) We plot the distribution of the policies sampled from the PCE surrogate model along a trajectory, where redder colours corresponds to policies with higher probability density under the surrogate model. This experiment uses 50,000 samples.

between learnt features, but crucially increases the Gaussianity of the latent space (we opt for $\beta = 4$). This allows us to approximate the latent space with a maximum-likelihood Gaussian distribution (with independent components) for which Hermite polynomials form an orthonormal basis. The $\beta-$VAE is trained using 500 random samples from the reward grid distribution. Note that we do not need to train a GFN for each one of these samples, as the latent representation is independent of the policy. For details on the implementation see App. E.

**Fitting a PCE with regularised regression.** Using any one of the models or an ensemble approach, we sample a trajectory through the discrete grid. In this case, we focus on the trajectory in Panel $b$), Fig. 1. For this given trajectory, we extract the policy at each step to yield a tensor $\boldsymbol{P} \in \mathbb{R}^{m \times n \times 5}$, where $m$ is the number of models in the ensemble, $n$ is the number of steps and 5 is the number of actions in the policy. Next, we construct an input-output dataset. For each of the $m$ models, we encode its associated reward grid to yield a 2-d Gaussian distribution, from which we can sample points. Each sampled point becomes an input that is mapped to the output tensor $\boldsymbol{P}$ of the associated model[6]. For each component of $\boldsymbol{P}$, we fit a separate degree 7 PCE model using ridge regression[7], using the logit transform (as described in Sec. 3.1 and App. B).

---

[6]This allows us to *augment* our training data to have more than $m$ data-points.

[7]Throughout, we opt for high degree PCE models, which are balanced via regularised regression. When using un-regularised regression, such expansions may be unstable. For a brief discussion on choosing PCE degrees and fitting methods, see App. B.

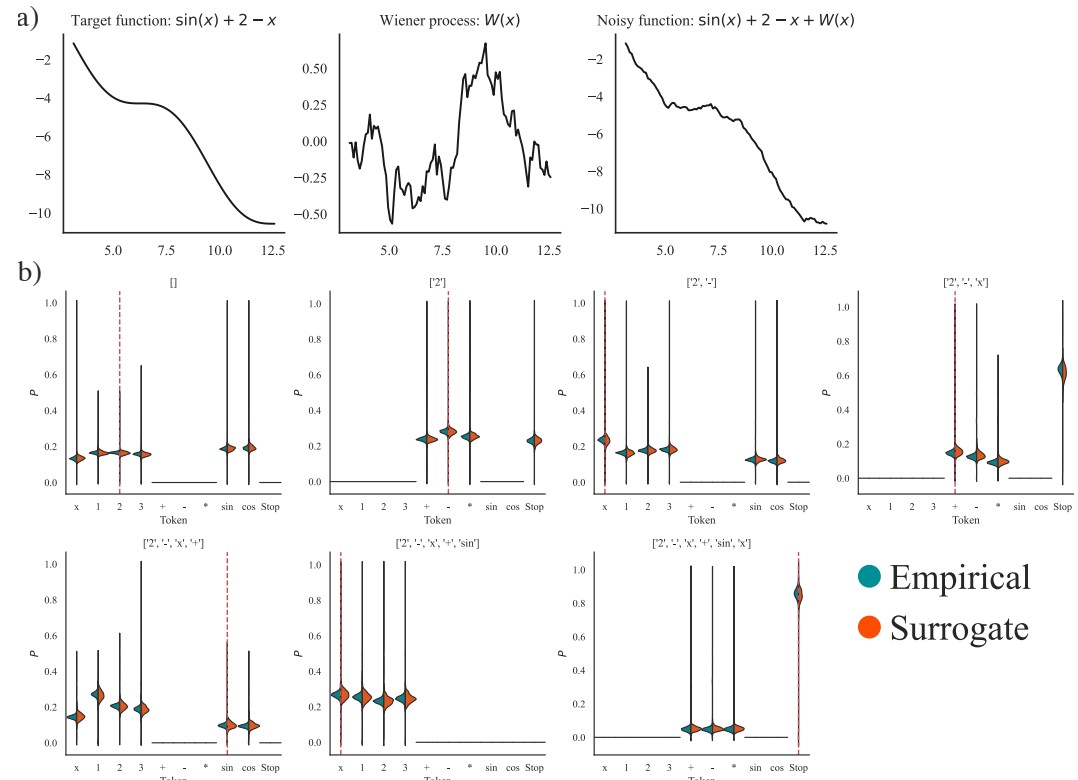

Figure 3: **Symbolic regression of a function with additive noise.** a) The target function $f(x) = \sin(x) + 2 - x$ in the range $[\pi, 4\pi]$, and a noisy function with additive Wiener noise. b) Empirical testing policies compared with surrogate policies from a PCE along a trajectory.

To perform MC sampling from the surrogate model, we first sample a reward grid, encode it to a latent Gaussian distribution, then sample a point from this distribution to serve as the input[8]. Panel $f$), Fig. 1 shows a comparison between 50,000 surrogate outputs[9] and an empirical testing ensemble of 100 models. We see that the surrogate model matches the distribution of the empirical samples. In particular, in the latter steps of the trajectory, when the model approaches the rewards, there is a bifurcation, where some models have a high termination probability, whilst for others it is more moderate. This differentiates between situations where the final position is a high or mid-reward spot. The surrogate model is able to capture this bimodality.

## 4.2 CONTINUOUS GRID WORLD WITH UNCERTAIN REWARDS

In this problem, we consider a continuous reward function defined on $\mathbb{R}^2$, that is defined by a mixture of two Gaussian distributions with means at $(x_1, y_1)$ and $(x_2, y_2)$ respectively, and with isotropic variance $\sigma^2 = 0.3$. As shown in Panel $a$), Fig. 2, starting at the origin, the model makes 5 steps, where the movements in the $x-$ and $y-$directions are sampled independently from two Gaussian distributions, which make up the policy. The reward is given by the position after 5 steps. We train a 'training' and 'testing' ensemble of models, where each is trained on a stochastically varied reward function. In particular, $(x_1, y_1) \sim \mathcal{N}((-1, -1), \sqrt{0.1}I)$ and $(x_2, y_2) \sim \mathcal{N}((1, 1), \sqrt{0.1}I)$, thereby slightly shifting the modes of the reward function, as shown in Panel $b$) of Fig. 2, leading to uncertainty in the policy. In this case, there is an obvious low-dimensional representation of each reward function, namely $\boldsymbol{\mu} = (x_1, y_2, x_2, y_2)$, which has a normal distribution. The policy at each step is described by $(\mu_x, \sigma_x^2, \mu_y, \sigma_y^2)$, the mean and variance of the Gaussian density in the $x-$ and

---

[8]Alternatively, we could sample from the Gaussian distribution which approximates the entire latent space, which we used to construct the orthonormal distribution — but this is an approximation.

[9]Even though there are only 625 reward grids, as each one is mapped to a distribution in latent space which can be sampled from, we can generate an arbitrary number of new samples.

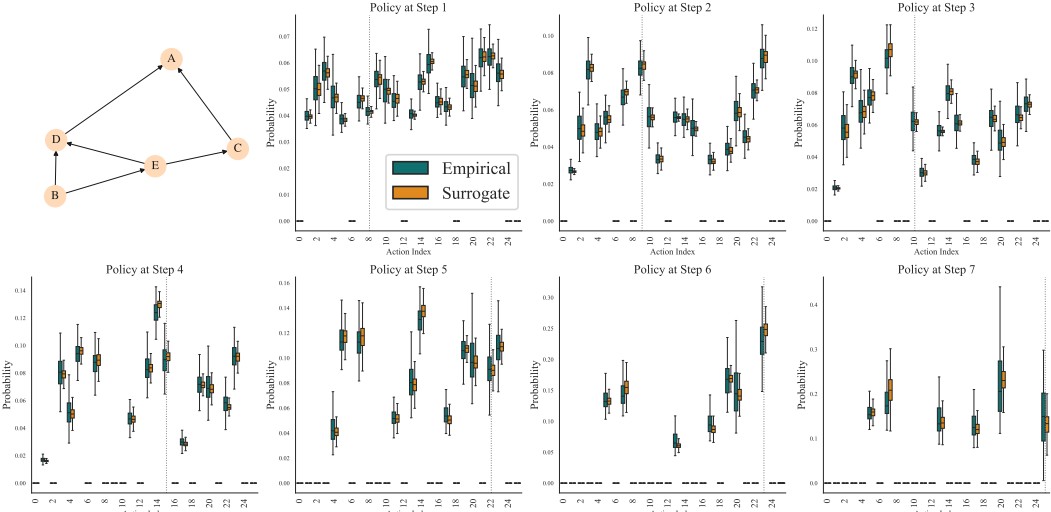

Figure 4: **Bayesian structure learning a linear Gaussian network**. *Top left*. The ground-truth Gaussian network. *Panels 2-8*. We show the policy, a distribution over 26 actions, for each of 7 steps along a trajectory. The policy distribution of the 250 empirical testing samples is shown in orange, whilst the 10,000 surrogate samples are in teal. The dotted line shows the action taken at each step. Outlier values beyond fences are not shown, but full scatters are presented in Fig. 6.

$y$−directions. We focus on a sampled trajectory (Panel $a$), Fig. 2), and extract the policy at each step. Using a basis of Hermite polynomials, we fit a PCE that maps between $\mu$ and the policy at each step, using degree 5 and ridge regression. Fig. 2 shows the distribution of the policy at each step, from both a testing ensemble of 100 GFNs (Panel $c$)) and 50,000 samples from the surrogate model (shown as a heatmap in Panel $d$)). The surrogate model can match the behaviour of the unseen testing samples, including the bimodality in the final steps, which emerges from each of the policies targeting one of the two modes.

### 4.3 SYMBOLIC REGRESSION OF A FUNCTION WITH ADDITIVE NOISE

In this next problem, we use a GFN to reconstruct the function $f(x) = \sin(x) + 2 - x$, from a small library of terms $\{x, 1, 2, 3, +, -, \times, \sin, \cos\}$, in the presence of additive Wiener noise, i.e. $\hat{f}(x) = f(x) + \sigma W(x)$, as shown in the Panel $a$) of Fig. 3. Each expression is built sequentially by adding a valid term at each step, with termination possible for a valid expression. The function is evaluated at 100 equispaced points between $[\pi, 4\pi]$ and then subject to additive noise. The reward function is given by,

$$R(g) = \frac{l}{1 + \frac{1}{100} \sum_{i=1}^{100} ||\hat{f}(x_i) - g(x_i)||^2}, \tag{7}$$

where $l = 1 + 0.2n$ is a bonus reward multiplier that encourages longer expressions, and $n$ is the length of the expression that forms $g$. In other words, the reward is inversely proportional to the mean-squared error between the data and the proposed function. For each realisation of the Wiener noise, this yields a slightly different reward function, implying epistemic uncertainty. To obtain a low-dimensional representation for each reward function from a dataset of $\{\hat{f}_j\}$, we first calculate the mean function from each sample and subtract it from our data, and then use the exact solution for the *Karhunen-Loève* expansion[10] of the resulting Wiener sample (see App. C) [Xiu (2010); Giambartolomei (2015)]. We take the first two components of the expansion $(z_1, z_2)$, which are normally distributed with $\mathcal{N}(0, 1)$, to be the low-dimensional representation of each reward function. We train a 'training' and 'testing' ensemble with 250 models each. Here we focus on

---

[10]An almost identical approach would be to use PCA on the set of function evaluations, but as we have the exact solution, we opt to use it.

the trajectory[11] $[2,—,x,+,\sin,x]$, and extract the policy from the training models, then fit a degree 14 PCE with ridge regression that takes $(z_1, z_2)$ as input, and outputs the logit of the probability distribution over tokens. The Panel $b$) of Fig. 3 shows the policy distribution of 150 testing models, compared with 10,000 surrogate policies, which match the empirical distribution closely.

## 4.4 BAYESIAN STRUCTURE LEARNING OF A LINEAR GAUSSIAN NETWORK

A Bayesian network (BN) is a DAG which shows the direct dependence relationships between random variables [Kitson et al. (2023)]. This is a general representation of a system that can be applied to a wide array of datasets in various scientific fields, helping to unravel the causal mechanisms within a system. However, reconstructing the dependence graph from finite observed data is a challenging and open problem. Deleu et al. (2022) present an approach for using GFNs to sample candidate BNs in proportion with their likelihood. They illustrate this on a linear Gaussian model using an Erdös-Rényi graph, as well as on flow cytometry data. The reward function is given by the *Bayesian Gaussian equivalent (BGe) score*, which computes the likelihood of the data given a candidate graph [Deleu et al. (2022); Kitson et al. (2023)]. As a result, each finite dataset yields an uncertain reward function that is sampled from the distribution $R \sim \mathcal{R}(\mathbb{E}[R])$. First, we train a 'training' and 'testing' ensemble with 250 GFNs each. Each model is trained with respect to a reward function calculated using the BGe score associated with a (different) dataset of 100 random samples from the (same) linear Gaussian network with 5 variables (see Fig. 4). To parametrise these reward functions in a low-dimensional space, we calculate the $r-$matrix[12]. This gives a 25-d vector that represents each reward function. We then perform PCA to extract the first two components, thus mapping each reward function to a point in 2-d. This distribution in $\mathbb{R}^2$ is well approximated by a Gaussian with independent components, which we estimate with MLE. Next, we consider a sequence of actions that constructs the ground-truth graph[13]. Each action adds an edge to the graph, with cycles being prohibited. We map between a (`source`, `target`) edge and an `action` using,

$$\texttt{action} = 5 * \texttt{source} + \texttt{target}, \tag{8}$$

where `source`, `target` $\in \{0, 1, 2, 3, 4\}$ and `action` $\in \{0, ..., 25\}$ with `action` $= 25$ corresponding to termination. We extract the policy at each step along the action sequence to yield a tensor $\boldsymbol{P} \in \mathbb{R}^{250 \times 7 \times 26}$, where 7 is the number of actions needed to build the ground-truth graph shown in the first panel of Fig. 4. For each of the $7 \times 26$ variables, we fit a degree 7 PCE with Hermite polynomials using ridge regression using the 250 training samples as the input-output data. We then sample 10,000 additional datasets from the linear Gaussian network, project them into the latent space, and use the PCE to predict the policy at these additional inputs. Fig. 4 shows the distribution of the policy for the 250 empirical testing samples and 10,000 surrogate samples for each of the 7 steps along the trajectory. Our surrogate model allows us to get a more complete estimate of the distribution of the policy stemming from uncertainty in the reward calculated from finite data.

## 5 DISCUSSION

GFNs learn a policy that has inherent epistemic uncertainty stemming from uncertainty in the reward function. Our UQ-framework leverages low-dimensional representations of reward function space to fit a PCE surrogate model and perform inexpensive MC sampling of policies along a specific trajectory. This allows for an understanding of the variations over policies that emerge from the noise and finitude of empirical data, which is essential when evaluating the predictions of a generative model. In future, such techniques could be used within the sampling procedure of GFNs to yield generative models that directly account for epistemic uncertainty. Moreover, our approach holds promise for more complex tasks, such as the generation of new molecular structures [Jain et al. (2023)], where prediction uncertainty has crucial implications. The use of polynomial models also carries the advantage that the surrogate has an interpretable functional form, allowing for sensitivity

---

[11]We choose this trajectory as it represents one way of constructing the exact solution out of many.

[12]The $r-$matrix is a Bayesian form of the covariance matrix associated with the BGe score [Geiger & Heckerman (1994)]. In this case, this is a $5 \times 5$ matrix that we flatten and then perform PCA.

[13]We could use any trajectory from the state-space DAG, but it is informative to see the policy along the desired solution. This is one trajectory that leads to the ground truth graph, but there are many equivalent ones, i.e. we can add the same edges in a different order.

analysis through the calculation of Sobol' indices [Sudret (2008)], however, we consider a comparison to a simple multi-layer perceptron (MLP) in App. F. Finally, whilst we have focused on GFNs here, such methods could be adapted for other generative models in the presence of uncertain inputs, such as next-token prediction with large language models (LLMs).

An anonymised repository with the code can be found here: `https://anonymous.4open.science/r/uq4gfn-518B/README.md`.

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

# APPENDIX

## A GENERATIVE FLOW NETWORKS

In Sec. 2.1, we briefly introduce the definition of a GFN. Here, we expand on this formulation, including specifying possible loss functions [Bengio et al. (2023)]. Previously, we defined the flow function, $F$, and introduced the flow matching condition, which we now derive more explicitly. In order for a function $F$ to define a valid flow network, it must satisfy,

$$F(A) = \sum_{\tau \in A} F(\tau), \tag{9}$$

for all $A \subseteq \mathcal{T}$ [Bengio et al. (2023)]. From this definition, we can define the *flow through a state* $s \in \mathcal{S}$,

$$F(s) := \sum_{\tau \in \mathcal{T}: s \in \tau} F(\tau), \tag{10}$$

and the *flow through an edge* $(s \to s') \in \mathcal{E}$,

$$F(s \to s') := \sum_{\tau \in \mathcal{T}: s \to s' \in \tau} F(\tau). \tag{11}$$

In other words, this enforces that flows are 'consistent', i.e. that the flow into a state is equal to the flow out of the state. As a consequence, the *flow matching condition*,

$$F(s) = \sum_{(s'' \to s) \in \mathcal{E}} F(s'' \to s) = \sum_{(s \to s') \in \mathcal{E}} F(s \to s'), \tag{12}$$

is satisfied for all $s \in \mathcal{S}$.

## A.1 LEARNING A FLOW APPROXIMATION

Given a DAG and a target reward function, constructing a valid flow is a non-trivial problem. We define a flow parametrisation of the pair $(G, R)$, which acts as a function approximator and seeks to minimise the error in some flow consistency condition. The simplest loss is derived directly from the *flow matching* condition in Eq. (1). The parameters are trained to minimise the loss,

$$\mathcal{L}_{\text{FM}}(s) = \left( \log \frac{\sum_{(s'' \to s) \in A} F_\theta(s'' \to s)}{\sum_{(s \to s') \in A} F_\theta(s \to s')} \right)^2, \tag{13}$$

which attempts to minimise the difference between the in-flow and out-flow at each state. As terminal nodes have no out-flow, the loss function is modified slightly such that the in-flow equals the reward [Bengio et al. (2021)].

A second choice is the *detailed balance loss* [Bengio et al. (2023)]. A valid flow satisfies the well-known detailed balance construct for Markov chains given by,

$$F(s)P_f(s'|s) = F(s')P_b(s|s'), \tag{14}$$

which can be used to define the loss function,

$$\mathcal{L}_{\text{DB}}(s, s') = \left( \log \frac{F_\theta(s)P_f(s'|s;\theta)}{F_\theta(s')P_b(s|s';\theta)} \right)^2, \tag{15}$$

again with a similar condition on the terminal nodes [Bengio et al. (2023)].

Third, we have the *trajectory balance* constraint, which enforces that for any complete trajectory $(s_0 \to s_1 \to ... \to s_n = x)$, a valid flow will have that,

$$Z \prod_{t=1}^{n} P_f(s_t|s_{t-1}) = F(s_n) \prod_{t=1}^{n} P_b(s_{t-1}|s_t), \tag{16}$$

where it is implicit that $P(s_n = x) = F(x)/Z$ [Malkin et al. (2022)]. For any trajectory $\tau = (s_0 \to s_1 \to ... \to s_n = x)$, we define the *trajectory balance loss* to be,

$$\mathcal{L}_{\text{TB}}(\tau) = \left( \log \frac{Z_\theta \prod_{t=1}^{n} P_f(s_t|s_{t-1};\theta)}{R(x) \prod_{t=1}^{n} P_b(s_{t-1}|s_t;\theta)} \right)^2, \tag{17}$$

where $Z_\theta$ is the estimate of the total flow given the learnt policy [Malkin et al. (2022)].

Finally, we have the *sub-trajectory balance* loss. For a valid flow, the trajectory balance constraint of Eq. (16) must hold, not only for a full trajectory, but for all partial trajectories as well [Madan et al. (2023)]. This constraint yields the objective,

$$\mathcal{L}_{\text{SubTB}}(\tau) = \left( \log \frac{F_\theta(s_m) \prod_{t=m}^{n} P_f(s_t|s_{t-1};\theta)}{F_\theta(s_N) \prod_{t=m}^{n} P_b(s_{t-1}|s_t;\theta)} \right)^2. \tag{18}$$

The policy can then be optimised over the parameters $\theta$ to minimise the chosen loss over a batch of sampled trajectories.

## B POLYNOMIAL CHAOS EXPANSIONS

In Sec. 3.1, we briefly introduce PCE, which we elaborate on here. First, we take the PCE defined previously, $Y = \sum_{j \in \mathbb{N}^m} c_{\mathbf{j}} \varphi_{\mathbf{j}}(\boldsymbol{X})$, where the $\varphi_{\mathbf{j}}$ satisfy the orthonormality condition,

$$\int_{\mathcal{X}} \varphi_{\mathbf{i}}(\mathbf{x}) \varphi_{\mathbf{j}}(\mathbf{x}) \rho_{\boldsymbol{X}}(\mathbf{x}) = \delta_{\mathbf{ij}}. \tag{19}$$

Here $\rho_{\boldsymbol{X}}$ is the density of the random vector $\boldsymbol{X}$ over support $\mathcal{X}$. In practical settings, it is necessary to truncate the infinite polynomial basis at a particular *degree*, $d$. We denote by $\Theta_{d,m}$ the subset of

$\mathbb{N}^m$ which corresponds to the numeration of terms in a truncation of the polynomial basis with $m$ inputs and maximum-degree $d$. Given input-output data, we fit a PCE using regression by solving the optimisation problem,

$$\{c_{\mathbf{j}}\} = \text{argmin}_{\tilde{c}_{\mathbf{j}}} \sum_{k=1}^{n} ||y_k - \sum_{\mathbf{j} \in \Theta_{d,m}} \tilde{c}_{\mathbf{j}} \varphi_{\mathbf{j}}(\mathbf{x}_k)||^2. \tag{20}$$

Regression does not impose constraints on the *collocation points*, $\{\mathbf{x}_k\}_{k=1}^{n}$, unlike pseudo-spectral projection methods [Xiu (2010)]. Additionally, the coefficients can be regularised with ridge of LASSO regression [Hastie et al. (2009)]. PCE is implemented using `ChaosPy` [Feinberg & Langtangen (2015)].

### B.1 MODELLING PROBABILITIES: THE LOGIT TRANSFORMATION

When the policy at each step is a discrete probability distribution over actions, this is represented by the vector $P \in [0,1]^m$ where $\sum_{i=1}^{m} P_i = 1$. As polynomial functions are not constrained to this range, we first transform $P$ using the logit transform,

$$\text{logit}(p) = \log\left(\frac{p}{1-p}\right), \tag{21}$$

which takes values in $\mathbb{R}$, with $\text{logit}(0) = -\infty$. We then fit the PCE to the transformed values. When taking samples from the PCE model, we undo the transform using the soft-max function to obtain a probability distribution over the $m$ actions.

### B.2 CHOOSING A POLYNOMIAL DEGREE

In the interest of brevity and due to lack of impact, we offer little explanation for the choices of polynomial degree in the PCEs. A higher-degree PCE includes more terms, which can prevent under-fitting and generate a more complete picture of the distribution. However, too high a degree can lead to the regression problem being underdetermined. We offer the following 'rule of thumb' for picking a polynomial degree. A polynomial of degree $p$ with $d$ input variables includes,

$$n = \binom{d+p}{p}, \tag{22}$$

terms. In order to solve the regression problem in Eq. (20), we need at least $n$ input-output samples to fit a $d-$dimensional PCE with degree $p$. Underdetermined regression problems can be solved with regularised regression [Hastie et al. (2009)].

## C KARHUNEN-LOÈVE EXPANSION

In Sec. 4.3, we use the KL expansion to embed noisy functions in a low-dimensional space. Here we provide the necessary mathematical background. A mean-zero stochastic process, $x(t)$, admits the decomposition,

$$x(t) = \sum_{k=1}^{\infty} z_k \sqrt{\lambda_k} \phi_k, \tag{23}$$

where $\lambda_k$ and $\phi_k$ solve the eigenvalue problem for the covariance operator [Xiu (2010)],

$$\int_0^T K(s,t)\phi_k(s) \, ds = \lambda_k \phi_k, \tag{24}$$

and $z_k \sim \mathcal{N}(0,1)$ are independent and identically distributed. For the Wiener process, this can be solved analytically to give [Giambartolomei (2015)],

$$\phi_k(t) = \sqrt{\frac{2}{T}} \sin\left(\frac{(k-\frac{1}{2})}{T}\right), \tag{25}$$

$$\lambda_k = \frac{T^2}{\left(\left((k-\frac{1}{2})\pi\right)\right)^2}, \tag{26}$$

where it is clear that the lower values of $k$ have the largest eigenvalues and therefore explain most of the variance. Using these expressions, it is possible to recover $\{z_k\}$ in an iterative fashion. For the symbolic regression task, we center the data by subtracting the mean, then assume the remaining signals are observations of the Wiener process. We then extract $(z_1, z_2)$ for each noisy function to obtain a low-dimensional representation of the reward. This approach is similar to, and intimately related to, PCA, except that it uses the analytical covariance operator for the Wiener process, as opposed to the empirical covariance.

## D   LINEAR GAUSSIAN NETWORK

In Sec. 4.4, we use the GFN model presented in Deleu et al. (2022) to demonstrate our approach. Here we describe the example problem in greater detail. We focus on the problem of sampling candidate graphs using data sampled from a linear Gaussian network.

A *linear Gaussian network* is a stochastic model defined by a causal DAG with adjacency matrix $(A_{ij})$. The model is defined by the relationship,

$$x_j = \sum_{x_i \in \text{Pa}(x_j)} \beta_{ij} x_i + \epsilon, \tag{27}$$

where $x_j$ is a random variable and $\text{Pa}(x_j)$ are the set of its *parent nodes*, i.e. the set of nodes $k$ such that $k \to j$ is in the DAG; $\beta_{ij} \sim \mathcal{N}(0, 1)$ if $A_{ij} = 1$ and 0 otherwise; and $\epsilon \sim \mathcal{N}(0, 0.01)$ is Gaussian noise [Deleu et al. (2022)]. Given a dataset $D$ and a candidate DAG, $G$, the reward is given by,

$$R(G) = P(G)P(D|G), \tag{28}$$

where $P(G)$ is the prior over DAGs, and $P(D|G)$ is the marginal likelihood of the observed data given $G$. Calculating the marginal likelihood is challenging; thus, we use the BGe score, under the assumption that the prior over both the parameters and structure of the DAG is *modular*. For further details, we direct the interested reader to Refs. [Deleu et al. (2022); Geiger & Heckerman (1994); Heckerman et al. (1995)]. In Sec. 4.4, we use a single underlying DAG with 5 nodes, shown in Fig. 4, and sample datasets with 100 points.

## E   ARCHITECTURES AND TRAINING

Here we describe the ML architectures used for the GFNs in Sec. 4.

### E.1   DISCRETE GRID-WORLD

#### E.1.1   $\beta$-VAE FOR LOW-DIMENSIONAL REPRESENTATIONS OF REWARD GRIDS

We represent each reward grid with a one-hot encoding over the three possible reward values. This is passed into a $\beta$-VAE where the encoder has a pair of convolutional layers, where the first has 3 input channels and 16 output channels, and the second has 16 input channels and 32 output channels. This is then passed through a linear layer to 128 units, and, finally, projected to a mean and log-variance in $d = 2$ dimensions. This is reversed in the decoder. We set $\beta = 4$, and train the model with Adam (learning rate, lr, set to lr = 0.001) for 1000 epochs. This is implemented in `Pyro` [Bingham et al. (2018)].

#### E.1.2   GFN FOR DISCRETE GRID

For this experiment, the GFN is parametrised by an MLP which takes a one-hot grid representation of the state as input, and passes it through two hidden layers with 128 units each, before a read-out layer with 5 outputs — corresponding to the 5 actions.

The model is trained to optimise the sub-trajectory loss defined in App. A.1, by sampling a number of trajectories to form a buffer, from which we perform priority trajectory sampling in proportion with their rewards [Schaul et al. (2016)]. In addition, we use $\epsilon-$greedy exploration with $\epsilon$ being

annealed during training, and soft-max temperature scaling to encourage discovery of high rewards. We use a batch-size of 64, Adam with an lr of 0.001, a temperature of 0.4, a maximum path length of 20, a buffer capacity of 10,000, and train the model for 20,000 episodes. The $\epsilon$-parameter is annealed with $\max(0.1, 0.5 * (0.99^k))$ where $k$ is the episode number.

### E.2 CONTINUOUS GRID-WORLD

The GFN for the continuous grid-world environment is parametrised using an MLP which takes a 3-d input, $(x, y, t)$, where the components are the $x-$ and $y-$ positions, and the step counter, respectively. The MLP has 2 hidden layers with 100 units each, and a 4-dimensional output layer corresponding to the Gaussian policy in the $x-$ and $y-$ directions, respectively, i.e. $(\mu_x, \log \sigma_x^2, \mu_y, \log \sigma_y^2)$ from Sec. 4.2. In this example, we train two MLPs with identical structures, one representing the forward model, and the other representing the backward model. The models are trained to optimise the trajectory balance loss. We train the model using Adam for 5000 episodes with a batch size of 256, and the minimum policy standard deviation 0.1 and a maximum of 1.

### E.3 SYMBOLIC REGRESSION

When performing the symbolic regression, we need to evaluate a function constructed from a sequence of tokens/terms from the library of terms specified in Sec. 4.3. We do so via conversion into *reverse Polish notation* (RPN) via the *Shunting-Yard algorithm* [Norvell (1999)]. The GFN that implements the symbolic regression environment embeds each token into a latent vector with 32 dimensions. The sequence of tokens is processed using an LSTM-RNN[14] [Hochreiter & Schmidhuber (1997)], where the final hidden state is the embedding of the whole expression. There are then three output heads: a forward model, which defines a probability distribution for the next token, a backward model, and the partition function. Expressions are capped at length 10.

This GFN is trained to optimise the trajectory balance loss for a batch of size 64, for 10,000 episodes with a learning rate of 0.001. We use temperature scaling in the soft-max with temperature = 1.5.

### E.4 BAYESIAN STRUCTURE LEARNING

In Sec. 4.4, we use the model published Deleu et al. (2022). Starting from the initial state $s_0$, a graph with no edges, graphs are constructed one edge at a time, masking actions that violate the DAG condition (see Appendix C of Ref. [Deleu et al. (2022)]). Each input graph is encoded as a set of edges, and each directed edge is embedded using an embedding for the source and target node, with an additional vector indicating the presence of edges in a graph $G$. The embeddings are passed through a linear transformer [Katharopoulos et al. (2020)] with two output heads, the first of which gives the forward transition probabilities over possible actions, and the second gives the probability to terminate the trajectory [Deleu et al. (2022)]. For further details see Deleu et al. (2022).

## F COMPARISON TO MULTILAYER PERCEPTRONS

In this section, we compare our PCE surrogate model to a simple MLP. Whilst they lack the analytical tractability of PCEs, MLPs have the advantage that they can fit multi-dimensional outputs simultaneously. In Figs. 5 & 6, we show the testing ensemble and PCE surrogate samples from Sec. 4, alongside surrogate samples from an MLP with two hidden layers, each with 64 units. As with the PCEs, we train the MLPs on a training ensemble of input-output data, where inputs are low-dimensional representations of reward functions and outputs are policies along a trajectory. For the discrete world, symbolic regression and structure learning task, this network is trained to minimise the Kullback-Leibler divergence between the surrogate and empirical distributions for the training ensemble. For the continuous grid world, the MSE is optimised instead. To illustrate that both surrogate models are able to capture the full breadth of the distribution, Figs. 5 & 6 show strip-plots rather than distributions. Whilst the MLPs are typically able to capture the relationship between

---

[14]Long Short Term Memory Recurrent Neural Network

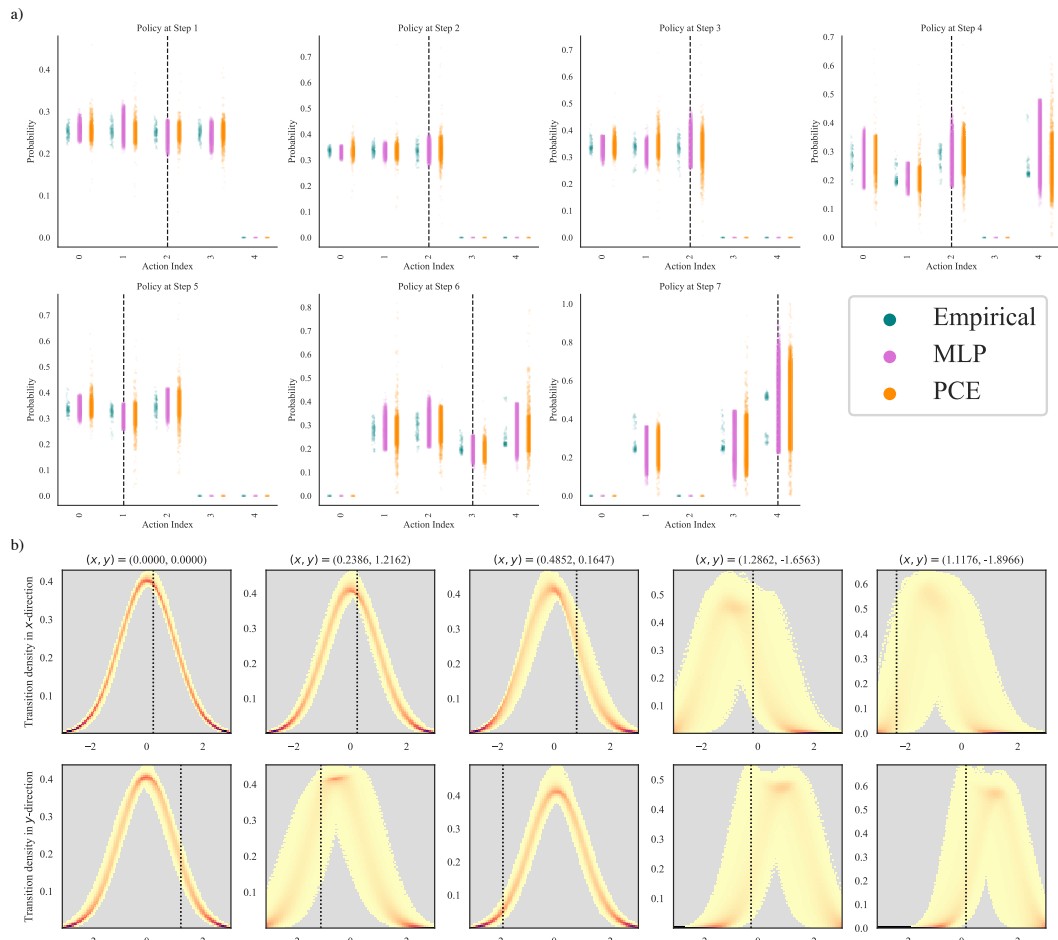

Figure 5: **Comparison to an MLP for grid-worlds.** $a$) Testing samples compared with surrogate samples from a PCE and MLP model for the discrete-grid world example from Sec. 4.1. We see that both surrogate models are able to capture the relationship between inputs and outputs, and that the surrogate model with its additional samples builds up a distribution with greater variance. $b$) Comparison for the continuous-grid world example from Sec. 4.2. The MLP can learn the input-output relationship, but suffers from regression to the mean, i.e. the surrogate model predicts values which are close to the mean, thus underestimating the variance of the empirical samples shown in Fig. 2.

inputs and outputs, in some cases they suffer from *regression to the mean*, where predicted outputs are often close to the mean value, thus underestimating the variance. This is most clearly illustrated in both examples in Fig. 6, and Panel $b$) of Fig. 5, when compared to Fig. 2. Finally, we note that MLPs do not allow for sensitivity analysis, such as the calculation of Sobol' indices [Sudret (2008)], which is a central feature of the PCE. The sensitivity analysis with NNs corresponds to examining the partial derivatives of the predictions with respect to the inputs. The Jacobian of the outputs with respect to the inputs evaluated at $x^*$ is given by,

$$J_{ij}(x^*) = \frac{\partial}{\partial x_i}\left(f_j(x^*)\right).$$ (29)

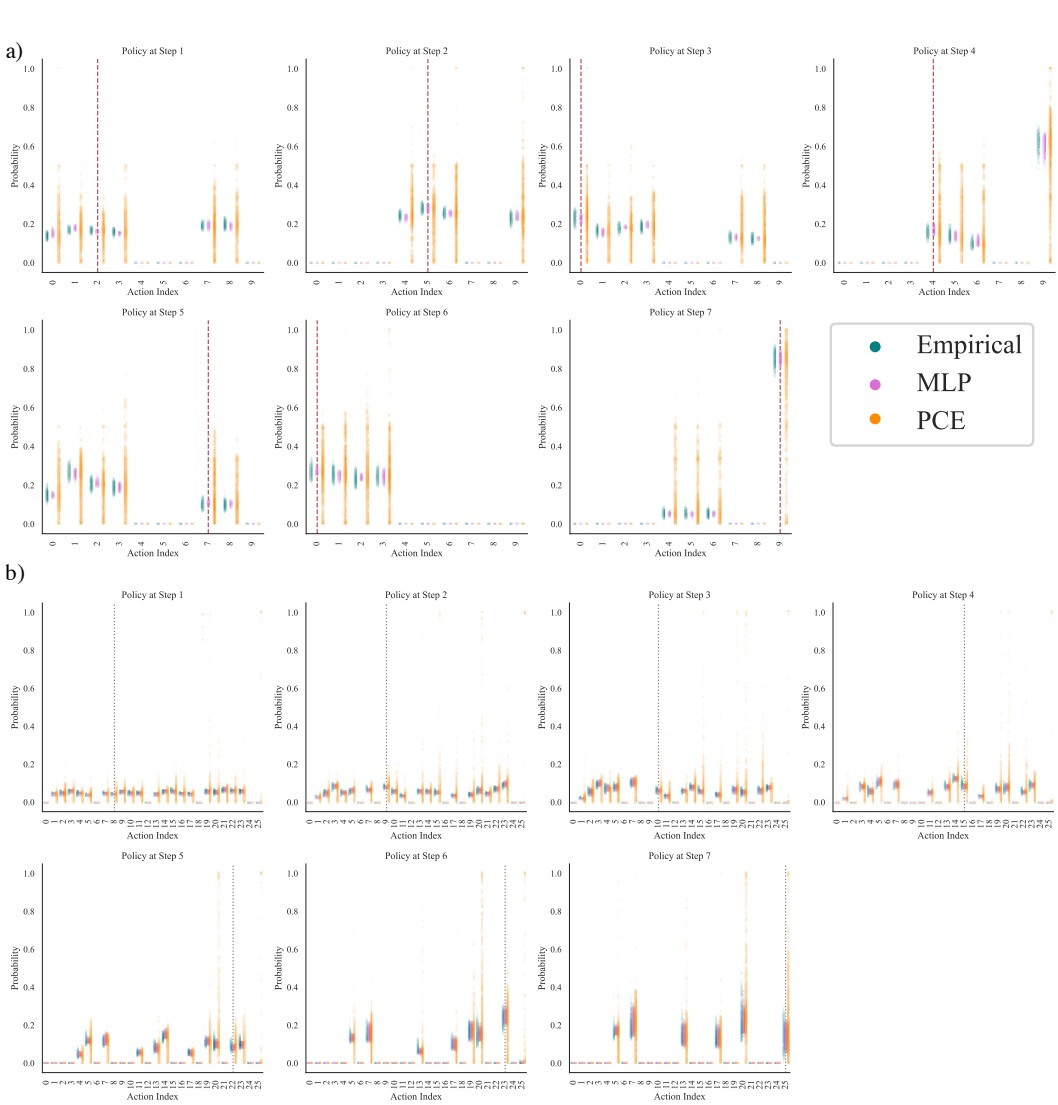

Figure 6: **Comparison to an MLP for symbolic regression and Bayesian structure learning.** $a$) Testing samples compared with surrogate samples from a PCE and MLP model for the symbolic regression example from Sec. 4.3. The MLP suffers from regression to the mean i.e. the surrogate model predicts values which are close to the mean, thus underestimating the variance of the empirical samples shown in Fig. 3, which does not occur for the PCE model. $b$) Comparison for the Bayesian structure learning example from Sec. 4.4. The MLP again suffers from regression to the mean.

