# OpenReview forum: "Surrogate-Based Quantification of Policy Uncertainty in Generative Flow Networks"
_ICLR.cc/2026/Conference — Submitted to ICLR 2026_

### Official Review · Reviewer_oohH · 2025-10-25

**Soundness:** 1
**Presentation:** 1
**Contribution:** 1
**Rating:** 2
**Confidence:** 3

**Summary:**

The authors proposed a way to estimate uncertainty for learned policies in GFlowNets, using a polynomial chaos expansion over the ensemble of already-trained flow networks to train a mapping from a latent reward-function space to a policy space. Using this linear model over the reward latent space, the authors estimated the policy uncertainty on several benchmarks, such as discrete and continuous grid-worlds, symbolic regression, and a Bayesian structure learning task.

**Strengths:**

- The authors addressed an interesting problem from a general GFlowNet problem;

**Weaknesses:**

- To my knowledge, GFlowNets typically address learning of the sampler in an **acyclic** environment, and making GFlowNet work in a non-acyclic environment requires many additional training tricks (see Brunswic et al. 2024, and Morovoz et al. 2025). The described grid-world environments are exactly non-acyclic, and thus, it is unclear how the model was trained.
- The paper lacks high-dimensional real-world motivational examples: the only not fully synthetic example is Bayesian structural learning, where the state space consists of about $10^4$ states (and even this example cannot be considered as a real-world example since it uses Erdos-Renyi random graphs and not real data). Thus, it is impossible to understand whether this method actually scales.
- Lack of a general mechanism to extract the reward features: for each environment, the authors use ad hoc approaches to do it. In particular, it also makes me doubt about the scaling abilities of this empirical approach.

## References

Brunswic, L., Li, Y., Xu, Y., Feng, Y., Jui, S., & Ma, L. (2024, March). A theory of non-acyclic generative flow networks. AAAI-2024.

Morozov, N., Maksimov, I., Tiapkin, D., & Samsonov, S. (2025). Revisiting non-acyclic gflownets in discrete environments. ICML-2025.

**Questions:**

- What is a motivation for using polynomial features instead of just a two-layer neural network over the latent space? The linear model lacks expressivity, even in the space of orthogonal polynomials (of course, if the function is smooth, any function can be estimated by a Fourier series, but I don't see any reason to do it there).
- What is the use of using the reward features as an additional feature for training the initial policies, as in temperature-conditional GFlowNets?
- How is it possible to use this uncertainty score to get an improved training or exploration for GFlowNets?

Kim, M., Ko, J., Yun, T., Zhang, D., Pan, L., Kim, W., ... & Bengio, Y. (2023). Learning to scale logits for temperature-conditional GFlowNets. ICML-2024.

---

> ### Author Response · Authors · 2025-11-19
>
> We thank the reviewer for their time and consideration.
>
> Weaknesses:
>
> - In the discrete grid world, we specify that we do not allow trajectories to go back on themselves. More generally, including in the continuous case we consider, the grid state, call it $x$, may have cycles. We avoid this very simply by introducing an augmented state $y = (x,t)$, where $t$ is the discrete time-step in the trajectory. As this only moves forward, it eliminates the possibility of a cycle in the state-space DAG. We note that this ‘trick’ is very well-known and standard (e.g. see Sec 3.3.1 of https://arxiv.org/pdf/2111.09266). We have added a comment on this in the paper.
>
> - We opt for a number of canonical examples for GFNs, e.g. grid worlds, symbolic regression and Bayesian structure learning. Whilst these may not be ‘real-world’ examples, they are proof of concept for our method, as is often the case for new methodologies. We note that the choice to use ER graphs rather than “real-data” does not impinge on the validity of the method, but was rather a choice made to align with Deleu et al’s previous work. Also, these problems allow us to have better control and understanding of the uncertainties that we aim to quantify.
>
> We note further that the continuous grid world contains an infinite number of states.
>
> Finally, we would comment that our approach is not constrained by the state space, as we fit a separate PCE model for each action at each step along a trajectory. Thus a more complex problem would simply require more PCEs rather than PCEs with any additional complexity. Adding a more complex example primarily challenges our ability to visualise the results, rather than confirming anything additional about our framework.
>
> - Whilst we do use different approaches to obtain the low-dimensional representation, this is in-keeping with how we would expect this method to be applied. Firstly we note that the standard approach should be to use a non-linear encoder, such as a VAE as we do in the first example. However, if there is an obvious low-dimensional representation, such as in the continuous example, then this is the best choice.
>
> We would comment that in the Bayesian structure learning and symbolic regression task, as the data is Gaussian and linear, PCA is the optimal representation. We found that if we used a VAE, it would find a representation (unsurprisingly) that was equivalent to the PCA, up to a linear transformation, thus it was unnecessary to introduce additional black-box models to the project.
>
> Questions:
>
>  - We note that any nonlinear function could be used to approximate the relationship. We compare our approach to an MLP in Appendix F, and argue that the MLP overfits. We would object to the reviewer’s use of the word ‘linear’ to characterise the PCE, which is an inherently nonlinear model. In the paper, we also motivate the use of PCEs which are standard in UQ as they allow for the analytical computation of Sobol indices, as well as stochastic moments directly from the polynomial coefficients. A choice of surrogate model has to be made, and each approach has their own merits and drawbacks. Here we happen to use PCEs, but that does not inherently change the message or merits of the paper.
>
> - This is a very interesting comment about how we could use our reward features to improve the training of a GFN. We will look into this and see if it can be included in future work. We thank the reviewer for this suggestion.
>
> - Again, we thank the reviewer for this suggestion. We note at the end of the paper, that an ultimate goal beyond this project, is to improve the sampling of GFlowNets by accounting for such epistemic uncertainty. We will endeavour to answer this question in future projects.
>
> We thank the reviewer again for their time and comments.

---

> > ### Comment · Reviewer_oohH · 2025-11-25
> >
> > I would like to thank the authors for their detailed response and thank you for resolving the concern about the cyclicity of the studied environments. However, my main concern regarding the lack of motivating examples remains unaddressed. As a result, the practical application and relevance of the proposed method are still somewhat unclear to me.
> >
> > In addition, since the latent spaces of nearly all the studied examples, except for purely synthetic hypergrid environments, appear to be linear, I remain uncertain about the generalizability of the method beyond these specific settings.
> >
> > Finally, it is still not entirely clear how this technique provides practical benefits. For instance, in Bayesian optimization, uncertainty regions are primarily helpful for addressing the exploration-exploitation trade-off. In other contexts, confidence intervals may be of limited interest unless they have formal validity (for example, in the sense of conformal prediction) or are directly tied to critical applications such as hallucination detection for LLMs.

---

### Official Review · Reviewer_Ta1f · 2025-10-28

**Soundness:** 2
**Presentation:** 3
**Contribution:** 2
**Rating:** 4
**Confidence:** 4

**Summary:**

The paper introduces a modeling framework for uncertainty quantification in Generative Flow Networks given the uncertainty in the reward function. The authors suggest to train a proxy model that maps the rewards (or low-dimensional reward representations) into the probability distributions over GFlowNet generation trajectories. To do this modeling step, he authors use a low-dimensional reward representation based on VAE and then fit a system of orthogonal polynomials (Hermite polynomials in most examples) in order to estimate the logits of GFlowNets forward probabilities. With this model, it is possible to estimate the uncertainly in the trajectory-wise generation probabilities without retraining the whole GFlowNet from scratch for each particular reward design. The authors demostrate their approach on a number of synthethic benchmarks, including hypergrids, continuous hypergrid, symbolic regression, and Bayesian structural learning (DAG generation).

**Strengths:**

The paper is well-written and addresses clearly the problem of uncertainty quantification of GFlowNet-induced policy. The sources of the problem (e.g., the reward model trained from the data) are clear, and the motivation for the study is clear.

**Weaknesses:**

1. While the paper is rather well written, I feel that its real applicability is rather limited, despite the claims of broader applicability (such as LLMs mentioned in conclusion). It is a classical statistical approach to aprpoximate the unknown mapping from rewards to policies with a polynomial basis, yet this method has its own drawbacks, as all methods in parametric statistics (the curse of dimensionality). This mapping can be non-smooth and complicated, especially in the claimed real-world applications, and the paper does not provide an argument why should we expect that such a mapping can be well approximated by a small-order polynomial.

2. The comparison between surrogate-predicted and ensemble-based policy distributions is visual and qualitative. No quantitative metrics such (KL divergence, total variation distance) are reported.

3.  Is it not clear that the mapping $\Lambda_t$ is well defined. It might be the case that there are more than 1 forward policies in gflownet, each one inducing the correct terminal objects distribution.

4. The authors did not provide theoretical analysis for their method.

Minor issues:

- The chosen notation with $\widehat{R}$ for the true expected reward is rather confusing for readers with statistical background;

- The polynomial chaos expansion in line 176 is a bit inaccurate, as $Y$ is expressed in this case not as a polynomial, but as a series of polynomials.

**Questions:**

1. Is it possible to consider the approach with more expressive functional family? It is not clear if the basis of polynomials is representative enough, especially beyond the Gaussian latents used in the examples. Is it possible to adopt RKHS or some ideas from Gaussian processes?

2. The polynomial-based regression is known to degrade quickly given the curse of dimensionality. How to choose the degree of approximating polynomial in more "real" applications? At first glance, reader would expect that this degree should quickly increase, making the proposed algorithm an interesting, yet impractical, approach.

3. Is the performance of the surrogate sensitive to the chosen latent representation (for example, for different VAE retraining or different latent dimension of the VAE)?

4. Are there any accessible metrics between the policy distribution estimated by the algorithm and the "exact" one, with full retraining of GFlowNet for every reward? One can measure some characteristics, such as total variation distance, Jensen-Shannon divergence, etc.

---

> ### Author Response · Authors · 2025-11-19
>
> We would like to thank the reviewer for their thorough review of our paper.
>
> Weaknesses:
>
> 1 - As such models are not analytically tractable, it is difficult to argue from a theoretical perspective that we can expect the mappings to be smooth. However, in practice, we are simply learning a relationship between a low-dimensional (2-4 dims) input and a single output. Our use of VAE naturally limits the curse of dimensionality. We find that our PCE models are stable and produce realistic outputs that appear to align with testing ensembles but with slightly greater variance, consistent with what one would expect. We note that it would be easy to extend our approach to use adaptive and multi-element PCEs (such as those defined in: https://www.sciencedirect.com/science/article/pii/S0021999105001919) which are able to handle non-smooth mappings.
>
> 2 -  Whilst we concede that we do not offer a quantitative metric of divergence, we note that, in the absence of a ground truth, these metrics would have no context and thus would be uninformative. Moreover, we note that obtaining a ground truth i.e. training a large ensemble of models with Monte Carlo sampling of rewards, would be unfeasible (hence the motivation for this framework). Without a true distribution to compare too, there would be no utility in offering a metric of divergence, as we discuss in the response in Reviewer 2 also.
>
> 3 - Here the reviewer makes an interesting observation. In theory, a GFlowNet can learn one of many possible ‘flows’ that solve the terminating distribution. However, in practice, we found that specifying an architecture and the specifics of the hyper parameters ‘break the tie’ causing the model to converge to the same policy given a reward function. For this reason, we specify in the paper that the GFN learns a flow “given a particular architecture and training objective”.
>
> 4 - We are unsure what type of theoretical analysis the reviewer is referring to here and which part of our method requires such an analysis. If the reviewer can provide more details, we would be happy to attempt to address this.
>
> We have changed our notation for the expected reward. We thank the reviewer for this comment.
>
> Writing a PCE in terms of the particular orthonormal basis is standard and far more informative than writing it as a sum of monomials, as the coefficients in this case are the ones that we estimate from the data.
>
> Questions:
>
> 1. There is a direct link between the standard forms of continuous probability distributions and the corresponding family of orthogonal polynomials. Thus, in the case of a non-Gaussian latent, we would choose the corresponding polynomial family. Moreover, our approach could be easily extended to use data-driven PCE, where the family is constructed without assumptions about the distribution (https://www.sciencedirect.com/science/article/pii/S0951832012000853).
>
> Whilst we had considered using an ‘uncertainty-aware’ approach such as Gaussian processes, these have a natural limitation. If a GP model predicts an output of high-variance, we cannot determine if this uncertainty arises because of the variability coming from the uncertainty in the reward function, or if the GP model itself is not effectively calibrated on this region of the input space. As a result, it becomes easy to confuse two fundamentally different notions of uncertainty in the same problem.
>
> 2. As a separate PCE is learnt for each action at each step, the ability of a single PCE to learn the mapping does not suffer from the curse of dimensionality. More complex problems with larger state-spaces and numbers of actions simply imply more PCEs, rather than more complex ones. As a result, we have no reason to expect the method to degrade as we transition to more complex settings.
>
> 3. This is an interesting point. Whilst it does not fundamentally change our framework, investigating the sensitivity of our approach to the dimension of the latent space, is an interesting future direction that we will consider.
>
> 4. We acknowledge that this would be a very informative experiment. However, the primary motivation for this project is that retraining a sufficiently large ensemble to generate the distribution is computationally prohibitive, as it would require the retraining of $10^5-10^6$ models with different reward functions. For this reason, we do not have a ground truth to directly compare our surrogate distributions. However, we note that if this was feasible, our approach would have little added value.
> With regards to the number of input parameters, we rely on nonlinear dimensionality reduction via a VAE to keep this small.
>
> With regards to the polynomial degree, it would be easy introduce some adaptive order refinement by checking the convergence of statistics, contribution to the total variance of different coefficients, and sensitivity of the surrogate to the increased order per dimension.
>
> We thank the reviewer for their time and their consideration.

---

### Official Review · Reviewer_CQPL · 2025-10-28

**Soundness:** 2
**Presentation:** 2
**Contribution:** 2
**Rating:** 2
**Confidence:** 3

**Summary:**

The paper presents an approach for uncertainty quantification in GFlowNets. The method learns a mapping from reward functions to a lower-dimensional latent space, after that learning a surrogate model based on polynomial chaos expansion from a small number of GFlowNets trained on given reward functions. The obtained model can be used estimate the uncertainty in the policy given uncertain rewards. The approach is evaluated on discrete and continuous grids, symbolic regression and bayesian structure learning tasks.

**Strengths:**

Up to my knowledge, the paper presents the first attempt at uncertainty quantification in GFlowNets, which is an interesting and important research direction. The presented methodology is highly novel in the context of GFlowNets.

**Weaknesses:**

I am struggling to understand the method presented in Section 3.2, which is, in my opinion, the central part of the paper. Do I understand correctly that a separate PCE must be learnt for each pair (state, action) in the environment? Why is there a sum over actions in the loss in the Equation 6? Are coefficients $c_j$ separate across different states and actions, or must they be the same? Why doesn't the predictive model formally depend on the state? If a separate PCE must indeed be learnt for each pair (state, action), I do not really understand the utility of the proposed approach since it cannot generalize between states.

Section 2.3 states that the object of interest is a collection of random variables corresponding to the forward transition probabilities in each state of the trajectory. However, I find this formulation to be a bit confusing since trajectories can have different lengths, thus the size of this collection would also be a random variable. Wouldn't it be more natural to state that the object of interest is the forward policy itself, i.e. a family of probability distributions over actions in each possible state?

In Section 3.1 it is stated that in the case when the output is a discrete probability distribution, logit transformation is applied to probabilities, a separate PCE is fit on each action, and then the result is transformed back into a distribution using softmax. However, applying logit transformation to each element of a probability distribution, and then applying softmax would generally give a different probability distribution than the one we started with. Because of this, I also find this part confusing.

Although the experimental evaluation is conducted on a number of different environments, all of them are either toy (discrete and continuous grids) or very small scale, as symbolic regression task has trajectories of very small length (5) containing symbols from a small set of terms (9), and bayesian structure learning task is limited to 5 variables. I believe that experiments larger in scale are crucial to demonstrate the practical utility of the proposed method.

In addition, line 426 states that a certain trajectory ([2,—,x,+,sin,x]) is considered in the experimental evaluation in the symbolic regression task. I find such experimental setup where only one trajectory is used to evaluate the approach too limited, as it is generally difficult to draw conclusions from the experiment when the model is tested on such a small set of objects.

Finally, I point out a relevant paper that considers a GFlowNet-like problem formulation with uncertain rewards, which the authors may find interesting [1].

References:\
[1] Jiralerspong et al. Discrete Compositional Generation via General Soft Operators and Robust Reinforcement Learning. 2025

**Questions:**

0) See Weaknesses.

1) It is stated that hypergrid task allows actions of moving left, right, up, and down (line 247). Wouldn't this result in an environment with cycles? The usual GFlowNet formulation works in acyclic DAG environments. There are works that consider GFlowNets in non-acyclic environments [1, 2], but then this should be additionally discussed as this is not a standard setting.

References:\
[1] Brunswic et al. A Theory of Non-Acyclic Generative Flow Networks. AAAI 2024\
[2] Morozov et al. Revisiting Non-Acyclic GFlowNets in Discrete Environments. ICML 2025

---

> ### Author Response · Authors · 2025-11-19
>
> We would like to thank the reviewer for their detailed review.
>
>  - In our method, we learn a distinct PCE surrogate model to map between a multidimensional input, which is low-dimensional representation of the reward, to a single output, which is a logit-transformed probability, which represents the probability of an action at a particular step in the trajectory. Therefore, for a complete trajectory, we would have to learn K different PCE models, where is K is the sum over all steps of the number of actions at each step. The PCE model is not meant to be a generalisable, intelligent model, that can be applied between states. Instead, it is a flexible, cheap and efficient model which can be applied many times over to obtain a better estimate of the variance in each output probability given a variation in the input. PCEs are also typically constrained to single-variable outputs.
>
>  - The reviewer correctly identifies a typo in Eq. 6. The sum should be over a missing index, which we now label $l$, and which corresponds to each model in the training ensemble. We hope that our framework is now clearer, and we thank you for identifying the typo.
>
>  - Depending on the specifics of the problem, the policy over actions at each state - which the reviewer refers to - is not constant over steps in the trajectory. For example, in the a grid environment with a fixed number of steps $L$, the policy at a particular square at time 1, and that same square at time $L-1$, is not the same.
>
> We would like to reiterate that our approach learns a separate PCE for each action and each step of the trajectory, thus the length of the trajectory or the number of actions that are possible do not change a single PCE model at all - they simply change the number of PCE models in total. Whilst this was seen as a constraint in the previous comment, hopefully it is clearer now why we have opted for this approach, as it eliminates problems with the exponentially large state-space, and with additional variables such as length of the trajectory. Moreover, these PCEs can be trained in parallel, thus improving the computational efficiency.
>
> - We thank the reviewer for this point. The softmax is not the direct inverse of the logit transformation, instead it is the sigmoid. However, applying the sigmoid to each of PCE output values separately does not guarantee that those separate outputs will sum to one (over the actions possible at a particular state), which would be required to be a discrete probability distribution, thus the softmax was one particular work around. In practice we found that this gave the same output as applying the sigmoid and then renormalising, thus we believed the approaches were approximately identical, and that renormalising was more ad-hoc.
>
> - We thank the reviewer for this point. Whilst, we agree that the experiments are not extremely large, we argue that this paper is more of a ‘proof-of-concept’ on a number of examples, with a range of complexity. In future work, this could be applied to molecular discovery or combinatorial optimisation problems that have also been solved with GFNs. As a more expansive problem only increases the number of PCEs rather than the complexity of each one, we have no reason to believe that our method would fail to scale effectively.
>
> - Our approach and analysis is naturally constrained to consider one trajectory at a time. In this case, we do not opt for a random trajectory, but instead, we look at the exact solution to the symbolic regression tasks. Looking at a number of different trajectories would complicate the visualisation of our results substantially, and would offer little in the way of additional validation of our framework.
>
> We thank the reviewer for pointing out a relevant paper, which we will read with interest.
>
> - In the discrete grid world, we specify that we do not allow trajectories to go back on themselves. More generally, including in the continuous case we consider, the grid state, call it $x$, may have cycles. We avoid this very simply by introducing an augmented state $y = (x,t)$, where t is the discrete time-step in the trajectory. As this only moves forward, it eliminates the possibility of a cycle in the state-space DAG. We note that this ‘trick’ is very well-known and standard (e.g. see Sec 3.3.1 of https://arxiv.org/pdf/2111.09266). We have added a comment on this in the paper.
>
> We thank the reviewer for their time and their consideration

---

### Official Review · Reviewer_ePsK · 2025-10-31

**Soundness:** 2
**Presentation:** 3
**Contribution:** 2
**Rating:** 2
**Confidence:** 3

**Summary:**

The paper introduces a surrogate modeling framework for uncertainty quantification in Generative Flow Networks under epistemic uncertainty in the reward function.
The authors train a small ensemble and fit a Polynomial Chaos Expansion surrogate instead of training multiple GFNs on perturbed rewards. The surrogate maps low-dimensional representations of rewards to policy statistics. This surrogate enables cheap Monte Carlo estimation of the marginal distribution over policies induced by uncertainty in the reward. Thus, retraining additional GFNs is not needed.
Experiments cover discrete grid, continuous grid, symbolic regression, and Bayesian network structure learning and show that the surrogate reproduces ensemble uncertainty at a reduced cost.

**Strengths:**

The paper addresses a clear point that Bayesian or Monte-Carlo ensembles for UQ in GFNs are computationally infeasible. The motivation of a surrogate-based hence is clear.
The separation between epistemic uncertainty in rewards and randomness in training (SGD, initialization) is stated clearly.
Experiments cover both discrete and continuous environments.

**Weaknesses:**

1.Line 125 is misleading: the distribution is built for policies, integrating out reward functions.
2. Though the idea to approximate the mapping with polynomials is appealing, this mapping can be highly non-smooth, discontinuous, and multimodal. No argument or empirical check supports that a low-order polynomial expansion provides a valid approximation.
3. The analysis assumes (Y=f(X)) (the policy statistics) have finite second moments, but no proposition proves that for general GFlowNet training. The PCE representation is mathematically invalid without bounded variance.
3. Authors choose the degree of the PCE arbitrary, no mathematical explanation provided. “lack of impact” as stated is not a justification. 4.Moreover, the influence of the approximation basis should be explored as an ablation. Convergence analysis is missing.
5. The comparison between surrogate-predicted and ensemble-based policy distributions is visual and qualitative. No metrics such as KL divergence, total variation distance, or correlation are reported.

**Questions:**

1. Should “marginal distribution over (\mathcal{R})” be “marginal over policies (\pi)” obtained by integrating over (\mathcal{R})? Please clarify and restate formally what distribution your method aims to estimate.
2. Under what conditions on the reward distribution and training process does the policy statistic (Y) have finite variance? Can you provide empirical evidence or theoretical reasoning supporting this assumption?
3. How is the polynomial degree selected? Why was no adaptive truncation, error estimator, or cross-validation procedure applied? What is the relationship between number of polynomial terms and GFN state space dimensionality?
5. Have you computed any quantitative divergence metrics (KL, JS, total variation, or correlation) between the surrogate-induced policy distribution and the ensemble reference? If not, how can readers assess the calibration or bias of the surrogate?
6. Why was Polynomial Chaos chosen over Gaussian Processes or kernel surrogates, which could model non-smooth mappings and yield built-in uncertainty estimates?
7. How sensitive is the surrogate performance to the chosen latent representation (PCA vs VAE)?
8. Do you have any empirical justification that the policy manifold in latent reward space is smooth enough for a low-order polynomial approximation?

**Details Of Ethics Concerns:**

-

---

> ### Author Response · Authors · 2025-11-19
>
> We would like to thank the reviewer for their thoughtful assessment of our submission.
>
> Weaknesses:
>
> 1 - see Q1; 2 - see Q7; 3 - see Q2; 3 -  see Q3; 4 - see Q3; 5 - see Q4
>
> Questions:
>
> 1. We believe you are referring to lines 143-145. Our method aims to calculate the variance in policy space that stems from variance in a reward function. In the revised draft, we have made this more clear. We are interested in the distribution over policies which can be obtained by integrating over all possible rewards. We have amended the text.
>
> 2. Here the reviewer offers a valuable insight. The original policy statistics (for the discrete examples) are probabilities between [0,1], which naturally have finite variance. However, we apply the logit transform which makes them unbounded. In order to guarantee bounded variance, we need that the policy statistic is in a compact sub interval [eps, 1-eps] for small epsilon, then its logit transform will also be bounded. For values outside this sub interval, we do not apply the surrogate model, as if all the training models learn that a particular action has probability 0 or 1, this stems from an action being enforced or disallowed at a particular state (thus the surrogate model copies this part of the policy). This can be seen throughout our discrete examples, where the surrogate model also has probability 0 for impossible actions. However, in more complex examples where some events occur with arbitrarily small probability, this should be addressed by direct clipping of probability values.
>
> 3. Polynomial degrees were chosen inline with the criterion given in App B.2. The relationship is not between the number of polynomial terms and the GFN state space dimensionality, but instead the number of models in the training ensemble, as this is the key quantity constraining the regression. This is because each policy output is fit with a separate PCE, thus the overall dimensionality of the GFN state space is not relevant to our approach.
>
> Whilst we did compare different polynomial degrees, we did not report these in the paper, as, in the absence of a ‘ground-truth’, there was no way to argue that a particular surrogate model was ‘better’ than another. Comparison between polynomial degrees does not change the framework, and adding in such comparisons would be graphically cumbersome, and uninformative.
>
> 4. Whilst we appreciate that our analysis is primarily visual, we would like to point out that providing a divergence measurement between the surrogate and testing data may not be very informative, hence was omitted. These metrics would be out of context in the absence of a genuine ground-truth (which cannot be computed without an extremely large Monte-Carlo sample - which is computationally prohibitive, a major motivation for the framework). These numbers on their own would not inform the reader of anything about the quality of the surrogate model. Moreover, we compare our surrogate output with a validation test distribution, and see strong agreement, despite the small sample size.
>
> 5.  The reviewer points out that PCE is just one possible choice of surrogate model. However, as we state in the paper, PCEs are chosen as they are a flexible and effective choice that allows for the calculation of important UQ quantities such as Sobol indices or stochastic moments directly from polynomial coefficients. Moreover, we do consider a comparison with an MLP in App F.
>
> Whilst we had considered using an ‘uncertainty-aware’ approach such as Gaussian processes (GP), it is worth noting that these have a natural limitation. If a GP model predicts an output of high-variance, we cannot determine if this uncertainty arises because of the variability coming from the uncertainty in the reward function, or if the GP model itself is not effectively calibrated on this region of the input space. As a result, it becomes easy to confuse two fundamentally different notions of uncertainty in the same problem.
>
> 6. We note that in the first example, we use a VAE because there is no reason to expect a linear dimensionality reduction to surpass a nonlinear representation. In the subsequent examples, we either use a direct, optimal low-dimensional representation or the problem is linear and Gaussian (in the symbolic regression and in the Bayesian structure learning), thus PCA is the optimal reduction. Moreover, we found that using a VAE on these problems gave a representation that was identical to PCA (up to a linear transformation).
>
> 7. As these models and their learned representations are somewhat intractable, we do not have direct evidence. However, it is clear that we learn stable PCE approximations that produce realistic surrogate output distributions. Moreover, we can easily extend our approach to use adaptive, multi element PCE that is able to handle non-smooth mappings (https://www.sciencedirect.com/science/article/pii/S0021999105001919)
>
> We thank the reviewer again for their comments and for their time.

---

### Meta-Review · Area_Chair_mxvv · 2026-01-02

**Summary:**

The paper addresses an interesting problem regarding uncertainty propagation (reward -> policy) in GFlowNets using a surrogate model based on polynomial chaos expansion. This is a novel take on uncertainty quantification in GFlowNets. However, the reviewers point out several issues regarding the design choices (the paper does not verify that the target mapping is sufficiently smooth to be approximated by a polynomial; the sensitivity to the choice of degree is not investigated in enough detail), the paper lacks quantitative evaluation, and the practical utility beyond small scale experiments is questionable.

**Reviewer Concerns:**

The concerns mentioned in the meta-review to motivate the recommendation largely remain

**Reviewer Scores:**

I don't think that they would have changed

---

### Decision · Program_Chairs · 2026-01-26

Reject